# Modeling Study on Optimizing Water and Nitrogen Management for Barley in Marginal Soils

**DOI:** 10.3390/plants14050704

**Published:** 2025-02-25

**Authors:** Renaldas Žydelis, Rafaella Chiarella, Lutz Weihermüller, Michael Herbst, Evelin Loit-Harro, Wieslaw Szulc, Peter Schröder, Virmantas Povilaitis, Michel Mench, Francois Rineau, Eugenija Bakšienė, Jonas Volungevičius, Beata Rutkowska, Arvydas Povilaitis

**Affiliations:** 1Institute of Agriculture, Lithuanian Research Centre for Agriculture and Forestry, 58344 Kėdainiai, Lithuania; virmantas.povilaitis@lammc.lt (V.P.); eugenija.baksiene@lammc.lt (E.B.); jonas.volungevicius@lammc.lt (J.V.); 2Department of Water Engineering, Vytautas Magnus University, 44248 Kaunas, Lithuania; arvydas.povilaitis@vdu.lt; 3Agrosphere Institute (IBG-3), Forschungszentrum Jülich GmbH, 52428 Jülich, Germany; r.chiarella@fz-juelich.de (R.C.); l.weihermueller@fz-juelich.de (L.W.); m.herbst@fz-juelich.de (M.H.); 4Field Crops and Plant Biology, Estonian University of Life Sciences, 51006 Tartu, Estonia; evelin.loit-harro@emu.ee; 5Institute of Agriculture, Warsaw University of Life Sciences—SGGW, 02-787 Warsaw, Poland; wieslaw_szulc@sggw.edu.pl (W.S.); beata_rutkowska@sggw.edu.pl (B.R.); 6Department Experimental Environmental Simulation, Helmholtz Center for Environmental Health, Neuherberg, 85764 Oberschleißheim, Germany; 7University of Bordeaux, INRAE, Biogeco, Bat B2, Allée G. St-Hilaire, F-33615 Pessac CEDEX, France; michel.mench@inrae.fr; 8Environmental Biology, Centre for Environmental Sciences, Hasselt University, 3590 Diepenbeek, Belgium; francois.rineau@uhasselt.be

**Keywords:** barley, crop modeling, marginal soil, nitrogen stress, water stress

## Abstract

Water and N availability are key factors limiting crop yield, particularly in marginal soils. This study evaluated the effects of water and N stress on barley grown in marginal soils using field trials and the AgroC model. Experiments from 2020 to 2022 in Lithuania with spring barley cv. KWS Fantex under two N fertilization treatments on sandy soil provided data for model parameterization. The AgroC model simulated barley growth to assess yield potential and yield gaps due to water and N stress. Potential grain yields (assuming no water or N stress) ranged from 4.8 to 6.02 t DW ha^−1^, with yield losses up to 54.4% assuming only N stress and 59.2% assuming only water stress, even with the N100 treatment (100 kg N ha^−1^ yr^−1^). A synthetic case study varying N fertilization from 0 to 200 kg N ha^−1^ yr^−1^ showed that increasing N still enhanced yield, but the optimal rate of 100–120 kg N ha^−1^ yr^−1^ depended on climatic conditions, leading to uncertainty in fertilization recommendations. This study underscores the importance of integrating advanced modeling techniques with sustainable agricultural practices to boost yield potential and resilience in marginal soils. Incorporating remote sensing data to capture soil and crop variability is recommended for improving simulation accuracy, contributing to sustainable agriculture strategies in the Baltic–Nordic region.

## 1. Introduction

Changing future climatic conditions is an urgent global challenge impacting agricultural systems and food production worldwide [1]. It is projected that by 2050 the world‘s population will surpass 9.5 billion, and consequently, food production needs to increase by 70–85% [2]. Given the scarcity of available arable land within Europe, the effective utilization of previously underutilized or marginal land (except those for biodiversity conservation) becomes a crucial strategy to keep production sufficiently high and even to increase production [3].

Fluctuations in climatic conditions during the growing season directly affect crop physiological processes, including growth, development, and finally, overall productivity in terms of total yield but also yield quality. Crop development mainly depends on air temperature and available water in the root zone, and therefore, fluctuations in these conditions directly affect the length of the growing season and overall growth [4]. Radiation, particularly sunlight, on the other hand, affects the assimilation rate by providing energy for photosynthesis [5], while the amount of precipitation and the water holding capacity of soil determine soil water availability for crop growth and transpiration [6]. Vogel et al. [7] highlighted that climatic factors occurring during the growing season at the global scale can elucidate 20–43% of maize, soybean, rice, and spring wheat yield variation, which is attributed to the explained climate-dependent variance. However, the impact severity of climate change on crop production varies between regions and crops.

Nitrogen (N) is essential for plant growth and development, significantly influencing yield formation patterns [8]. At the same time, N is the most limiting macronutrient for crop production in most agricultural areas worldwide. Consequently, farmers often tend to overfertilize with N to ensure maximum yields and avoid yield reductions associated with N deficiency [9]. In addition, the overall efficient use rate of N fertilizers is usually around 50%, meaning the remaining unused N potentially leaches after crop harvest [10]. Continuous overfertilization of crops without considering plant needs has led to nutrient pollution becoming a major environmental problem for groundwater and surface waterbodies [11].

Barley (*Hordeum vulgare* L.) holds a significant position among cereals, serving an essential role in both animal feeding and human consumption (e.g., in malting) [12]. Barley stands as one of the most ancient, cultivated crops worldwide, and due to its adaptability to various climatic conditions, it can be grown in highly productive high-input agricultural systems but also in subsistence or low-input agriculture, making it a versatile and resilient crop [13]. In Europe, over the past 5 years, the total barley cultivation area remained relatively constant, varying in the range of 22.5–24.2 million ha^−1^. Hereby, slightly more than half (54.8–57.6%) of the barley cultivation area is concentrated in Eastern Europe, while Northern, Southern, and Western Europe have similar barley cultivation shares, accounting for 13.8–14.4%, 13.8–14.3%, and 15.3–16.3%, respectively [14]. Despite the fact that Eastern Europe accounts for the major part of the barley area, the grain yields in this region are the lowest with 2.6–3.1 t ha^−1^. A similar yield, i.e., 2.5–4.1 t ha^−1^, is obtained in Southern Europe, while in Northern and Western Europe, barley yield is higher with approximately 4.3–5.9 and 5.8–6.9 t ha^−1^ [14]. Northern Europe is generally well suited for barley cultivation due to its favorable climate and fertile soils. However, at present, the main limitations of barley growing in Northern Europe are the occurrence of drought, rain during sowing and harvest, and heat stress [15]. According to the global yield gap atlas (www.yieldgap.org), the barley yield potential in Northern Europe varies between 6.4 and 7.2 t ha^−1^, while the relative yield gap (difference between actual and yield potential) is 35.3–62.9%, respectively.

Despite barley being a rapidly maturing and relatively drought-tolerant crop, the relative yield gap is likely to be much higher when cultivated on marginal soils compared to cultivation on fertile soils. In this context, marginal soils can be defined by their biological, physical, economic, and environmental limitations or constraints for agricultural production. For instance, it refers to land that does not permit sustainable food production or normal utilization [16]. According to the Muencheberg Soil Quality Rating (SQR) system, 46% of European land is classified as “marginal” [17]. In addition, according to the EUSO soil health dashboard, 61% of EU soils are in an unhealthy state. Based on both sources, the field experiment at the barley growth site described below falls within the poorest area of Lithuanian soils, where sandy soil textures prevail. As a consequence, these soils have low water and nutrient retention and are also characterized as acidic soils. Therefore, as the global population grows and pressure on productive agricultural land increases, there is a demand to bring back more non-fertile soils into agricultural use. Despite the favorable conditions for barley cultivation under Baltic–Nordic environments, there is still a lack of experimental data, especially regarding yield potential and yield gaps when barley is grown under marginal soil conditions. The results of such a study could be highly beneficial in providing guidance for barley breeding tailored to low-fertility soil conditions.

Obviously crop growth, biomass, and yield development are significantly influenced by prevailing weather and soil conditions, as well as by management practices, such as sowing date, seeding density, and fertilizer application [18]. Under field conditions, identifying these limiting factors (such as the interaction between N and water stress) on plant growth and yield is challenging, especially when analyzing barley production in marginal soils. However, the complex relationships among these factors are well captured in state-of-the-art process-based models, and therefore, such models can serve as vital decision-support tools for farmers and farm advisors, enabling the calculation of potential crop yields and yield gaps under current or even future climate conditions [19,20] but also under various management strategies, such as different fertilization levels.

As outlined, there is still a lack of studies analyzing the yield potential and yield gaps of barley production on marginal soils under Baltic–Nordic conditions. To address these issues, the process-based agroecosystem model AgroC [21,22] based on a multiyear barley experiment with contrasting climatic conditions was calibrated and validated, and subsequently used to simulate barley growth responses to various N fertilization scenarios. The AgroC model was selected for its physically based soil water module, which may offer superior performance compared to traditional water bucket models commonly used in crop growth simulations, particularly for modeling barley growth under nitrogen and water stress conditions

The objectives of this study were as follows: (1) calibrate and evaluate the performance of the AgroC model using three-years’ field experimental data from barley cultivation on a marginal soil in the Nemoral climate zone; (2) to assess the potential spring barley yield and to untangle and quantify the confounding impacts of N and water stress, and (3) to unravel the impacts of increasing N application rates on barley yield, N uptake, nitrate leaching, and the intensity of abiotic stress.

## 2. Results and Discussion

### 2.1. Field Observations

The barley growing period lasted 108 days in 2020, 96 days in 2021, and 98 days in 2022. The total aboveground barley biomass (TAB) yield during this period varied significantly, from 4.15 ± 0.49 t ha^−1^ in 2020 to 4.97 ± 0.68 t ha^−1^ in 2021 for N100 treatment plots. In contrast, the TAB yield for the N0 (non-fertilized) treatment plots was approximately half of that found for the N100 plots, ranging from 1.82 ± 0.21 t ha^−1^ in 2020 to 2.49 ± 0.70 t ha^−1^ in 2021. These results indicate substantial differences between the growing seasons and fertilization conditions. From sowing to harvest, the mean air temperatures were 15.9 °C in 2020, 18.6 °C in 2021, and 17.9 °C in 2022 (hence 0.3 °C lower, 1.9 °C higher, and 1.7 °C higher compared to the 1990–2020 long-term average, respectively). Precipitation amounts during the barley growing seasons were also very variable. In 2020, the total precipitation amount was only 225.4 mm (94.7% of the long-term average), while in 2021 and 2022, similar amounts were recorded, with 327.8 (147.7% of the long-term average) and 346.2 mm (145.5% of the long-term average), respectively. Consequently, the 2020 season will be referred to as “normal” for both temperature and precipitation regimes, while 2021 and 2022 are referred to as “hot” and “wet”.

According to the official statistics in Lithuania [23], the average yield of spring barley across the country was 4.22, 3.30, and 3.81 t ha^−1^ in 2020, 2021, and 2022, respectively. These data were used to estimate the difference between the barley yields obtained on farmers’ fields and those from the field experiments with the N100 fertilization rate. The barley yields from the field experiments were lower than the average yield achieved across the country by 50, 33, and 41% in 2020, 2021, and 2022, respectively. These differences are mainly caused by the marginal soils the barley grew on in the experiments. In contrast, when comparing the yields from the field experiments with those from the same area (e.g., around Vilnius region), where predominantly similar soil types (Arenosols) were reported, the yield was only 20–30% lower in 2020–2022. This difference was likely the result of slightly lower fertilization rates used in our experiment compared to those typically used by farmers in the region.

### 2.2. Model Calibration and Validation

#### Soil Water Content

The measured soil water content (SWC) across five depths during the three year experimental period and for the N100 plot is illustrated in Figure 1. Additionally, statistical indicators, such as d, RMSE, and BIAS, are detailed in Table 1. Changes in measured SWC were higher in the arable Ap (0–30 cm) and B1 (30–50 cm) soil horizon compared to deeper horizons, such as B2 (50–78 cm), 2C(k) (78–105 cm), and 2C (105–120 cm). During the barley growing season, SWC values at 15 cm depth varied within a range of 0.05–0.29 cm^3^ cm^−3^, while at 40 cm depth, SWCs varied from 0.06 to 0.18 cm^3^ cm^−3^. At a larger depth, the SWC ranged even more narrowly, and the soil profile was never fully saturated or even near saturation. However, in the year 2020, which was referred to as “normal”, the SWC values across all soil horizons were lower and exhibited less fluctuation in relation to the hot/wet years of 2021–2022. Despite the varying trends in SWC fluctuations, the AgroC model matched the SWC dynamics well after calibration for the experimental years. Slightly better statistical outcomes were achieved during the calibration period (2020 and 2022) compared to the validation period (2021). The agreement between simulated and measured SWC at a 15 cm depth was reasonably good and similar for both the calibration and validation periods, with d values of 0.620 and 0.788, RMSE of 0.051 and 0.067 cm^3^ cm^−3^, and BIAS of 0.066 and 0.040, respectively. At the B1 horizon at 40 cm depth, the correspondence between measured and simulated SWC was acceptable, showing slightly better performance during the calibration period (d = 0.542, RMSE = 0.031 cm^3^ cm^−3^, BIAS = −0.001) compared to the validation period (d = 0.391, RMSE = 0.039 cm^3^ cm^−3^, BIAS = −0.006). In contrast, at 90 cm depth, the SWC values were underestimated for both periods (calibration and validation), particularly for 2021 and 2022 (Figure 1). At the deepest measured depth for SWC (110 cm), which corresponded to the lowest rooting depth of the barley, the SWC values varied the least, and the agreement between measured and simulated values was fairly good as well. For the calibration period, the average d index was 0.426, with a small RMSE of 0.015 cm^3^ cm^−3^ and a BIAS close to zero at 0.011. For the validation period, the corresponding values were similar with a d index of 0.455, RMSE of 0.008 cm^3^ cm^−3^, and a BIAS of −0.003.

### 2.3. Leaf Area Index and Barley Development Stages

The comparisons between simulated and measured barley development stages (DVS) after calibrating the phenology are presented in Figure 2. Throughout the years 2020–2022, crop germination, aided by favorable weather conditions, typically spanned 7–10 days until emergence. The barley flowering (BBCH = 60–69 or DVS = 1) and physiological maturity (BBCH = 92–99 or DVS = 2) stages were recorded 60–68 and 95–107 days after seeding, respectively. The corresponding simulated values for the 2020–2022 period for flowering and physiological maturity varied between 56 and 68 and between 92 and 104 days after seeding, indicating a strong concordance between the simulated and measured DVS. During the barley growing seasons from 2020 to 2022, only a few days of crop phenology differences were observed between the N100 and N0 treatments. Crop phenology is primarily determined by genetic characteristics and environmental factors, such as temperature, rainfall, and photoperiod [24]. However, the slight differences in crop development observed over those few days were likely influenced by the varying nutrient conditions.

During the model calibration (2020 and 2022) and validation (2021) periods, there was a close match between the simulated and observed crop leaf area index (LAI) (Figure 2, Table 1). During calibration, the simulated LAI values for the N100 treatment were slightly overestimated, as indicated by a d index of 0.892, an RMSE of 0.482 m^2^ m^−2^, and a BIAS of 0.291. However, for the validation period, the simulated results were closely aligned with the measured values, demonstrating a high d index of 0.925, a RMSE of 0.875 m^2^ m^−2^, and a BIAS close to 0 at 0.037. For the N0 treatment, the simulated LAI values were slightly underestimated but fell within the standard deviation limits. During the calibration period, the d index was 0.793, with a RMSE of 0.524 m^2^ m^−2^ and a BIAS of −0.484. Similar LAI statistical results were observed for the validation period, with a d index of 0.719, a RMSE of 0.795 m^2^ m^−2^, and a BIAS of −0.787. Here, it has to be noted that the mismatch in model prediction for the individual years were likely caused by different plot locations within the field as the soils in the field were not fully homogeneous and even small differences in soil texture or bulk density will have a large impact in the soil hydraulic characteristics of the extreme coarse and well sorted sandy soil.

### 2.4. Partitioning of Total Above-Ground Biomass, Leaf, Stem, and Grain Yield

The measured and simulated total aboveground biomass (TAB) as well as biomass of the individual organs for two contrasting N treatments, N0 and N100, are presented in Figure 2 and Table 1. The statistical measures showed that for the N100 treatment, the model slightly better reproduced seasonal variation of the TAB and its individual components during the calibration period compared to the validation period. For N0 treatment, the results were significantly better during the validation period, with d values very close to 1.

For example, for the N100 treatment, during the calibration period, the d value was high at 0.976, associated with a low RMSE of 0.475 t ha^−1^ and BIAS = 0.082 for TAB, but the d value (=0.925) decreased for the validation period in terms of RMSE = 1.197 t ha^−1^ and BIAS = −0.973. Analyzing the grain yields for the N100 treatment, the trends were similar to the observations for TAB. However, during the validation period, there was a slight overestimation (see Table 1) of biomass and grain yield (GY). These differences in simulated TAB and GY for the validation period likely resulted from changes in grain components, specifically the grain number and 1000 grain weight. In 2021, there was a larger grain count but lower grain weight.

Specifically, during the calibration period, the weight of 1000 grains ranged from 40 to 47 g, whereas during the validation period, it was only 30–35 g. Additionally, variations in the grain nutrient content were observed. During the calibration period, the N content in barley grain was approximately 2.36%, compared to a lower N content of about 2.08% during the validation period. Despite the high N100 fertilization rate, this indicated greater N stress in 2021 (validation period) compared to the calibration periods of 2020 and 2022. On the other hand, the P and K contents in both stems and leaves remained similar throughout the 2020–2022 period.

As indicated by the statistical indices listed in Table 1, the AgroC model showed varying degrees of accuracy in simulating leaf and stem biomass for the N100 treatment. For example, for stem biomass, better statistical indices were obtained during the calibration period with a d index of 0.790, RMSE of 0.570 t ha^−1^, and BIAS of −0.176, compared to the validation period with a d index of 0.570, RMSE of 0.928 t ha^−1^, and BIAS of −0.786. Conversely, for leaf biomass simulation, the model showed better predictions during the validation period compared to the calibration period.

The AgroC model performed exceptionally well in simulating TAB for the N0 treatment during the validation period, with very good statistical measures. This contrasted with the calibration period, in which the performance was not as strong, particularly in 2022. During the calibration period, the d index was 0.806, RMSE was 0.450 t ha^−1^, and BIAS was 0.314. In comparison, the validation period showed a high d index of 0.998, a very low RMSE of 0.051 t ha^−1^, and BIAS of 0.034, indicating almost perfect model accuracy. Similar trends were observed when analyzing the dynamics of storage organs for the N0 treatment. The model performed better during the validation period compared to the calibration period. During the calibration period, the model simulated storage organs with moderate agreement, a d index of 0.567, RMSE of 0.326 t ha^−1^, and BIAS of 0.262. In the validation period, these measures improved significantly to a d index of 0.975, RMSE of 0.166 t ha^−1^, and BIAS of 0.027,. Conversely, the AgroC model performed better in simulating leaf and stem biomass in unfertilized plots during the calibration period compared to the validation period. There was a noticeable increase in RMSE and a shift from overestimation to underestimation during the validation period.

Overall, the accuracy of the AgroC model during calibration and validation should be discussed with the understanding that this model was designed to predict crop yields at the field and field plot scale (point simulation). As such, the field was assumed to be uniform, without accounting for spatial differences in crop development caused by small scale changes in soil characteristics and/or crop management. Calibration and validation results for both N0 and N100 treatments indicated that while the model was highly effective, some underestimations and overestimations of biomass and individual organs may be due to changes in N availability under different environmental conditions. Moreover, sandy soils (where the field experiments were conducted) may exhibit significant morphological, chemical, and physical variability at various scales [25] even though they look fairly homogeneous on the other side. Previous studies have shown that soil parameters, such as soil organic carbon (SOC), but also soil texture, such as clay and sand content, can vary by more than twofold even within a small 3–4 ha field [26]. Therefore, the spatial variability of soil parameters could likely influence the model results in our study as the plots used in the three years were not the same. These factors could contribute to discrepancies between measured and simulated values, emphasizing the importance of considering soil heterogeneity and environmental variability in future model predictions even for apparently homogeneous sandy soils used in this study.

### 2.5. Estimated Water Stress

Despite barley being considered relatively tolerant to abiotic stress compared to other cereal crops [27], specifically water stress might be an important growth limitation as the field experiments in this study were performed on a highly sandy Arenosol soil (refer to Appendix A) with a low water retention capacity. Therefore, the simulated water stress over the three contrasting growing seasons was analyzed and is presented in Figure 3. The average water availability over the entire root zone, α_avg_ (-), which ranges from 1 to 0, was used to identify periods of water stress. α_avg_ is thereby the ratio of actual to potential transpiration and values of 1 indicate no water stress, while smaller values indicate water stress. Based on the plots for the years 2020–2022, the occurrence of water stress and water availability reflected the precipitation and potential evapotranspiration distribution over the growing seasons. The cumulative potential evapotranspiration from 2020 to 2022 was approximately 390–400 mm during the growing season, while the precipitation amount varied between 225.4 and 366.5 mm, indicating an insufficient water supply for barley growth. Despite the different fertilization treatments (N0 and N100) and associated differences in biomass production, the water stress experienced was nearly identical for both treatments. For example, in 2020, the average water stress α_avg_ for the entire barley growing season was 0.76 for the N0 treatment and only slightly smaller with 0.77 for the N100 treatment. In contrast, in 2021, α_avg_ showed higher values for both treatments with 0.86 for N0 and 0.84 for N100, indicating lower water stress compared to 2020. In 2022, α_avg_ values were 0.75 for N0 and 0.74 for N100, indicating the largest stress across all 3 years.

When analyzing specific barley growth periods separately, namely the vegetative stage (from sowing to flowering) and the reproductive stage (from flowering to physiological maturity), it was observed that the average water stress during the vegetative period was fairly consistent across all years. The α_avg_ values during the vegetative period were as follows: in 2020, N0—0.86 and N100—0.85; in 2021, N0—0.80 and N100—0.80; and in 2022, N0—0.86 and N100—0.86. However, the reproductive period experienced more severe water stress in 2020 (α_avg_ values of 0.63 and 0.64 for N0 and N100, respectively) and in 2022 (α_avg_ values of 0.48 and 0.52 for N0 and N100, respectively). In contrast, the reproductive period in 2021 showed much less water stress, with average α_avg_ values ranging from 0.91 to 0.95 for both treatments. It is also important to note that the intensity of α_avg_ reached its peak at different times in different years. For example, in 2020, the highest α_avg_ was simulated during multiple periods, reaching severe stress conditions with an α_avg_ of 0.2–0.3 during the stem elongation–booting growth stage (BBCH31–BBCH47) and even dropping to 0.08 during the flowering–development of grain growth stage (BBCH61–BBCH71). In 2021, the highest α_avg_ was simulated only during the leaf development stage, up to BBCH13, where water stress ranged between 0.36 and 0.44. In 2022, the highest α_avg_ occurred during the stem elongation stage (0.21–0.35) and the flowering–development of grain stages (0.17–0.36).

According to the FAO [14], barley’s water demand per growing period typically ranges between 400 and 650 mm, with the exact amount largely depending on the length of the growing season, environmental conditions, soil type, and the genetic characteristics of the variety. During the germination and early growth stages (BBCH stages), barley requires approximately 50–100 mm of water, accounting for about 15.4% of the total seasonal water requirement. As the crop enters the vegetative period, from tillering to stem elongation, water demand increases, requiring an additional 100–200 mm, which represents around 30.8% of the total water need. The most critical phase is the reproductive period, particularly from heading to grain filling, where the water requirement peaks at 200–250 mm, making up about 38.4% of the total. Finally, during the maturation period, barley needs approximately 50–100 mm, accounting for the remaining 15.4% of the total water demand. Throughout the growing seasons, under rainfed conditions, the crops consistently experienced water stress due to insufficient water supply in our experiments. However, the intensity of this stress varied from mild to severe. As a result, farmers cultivating crops on marginal soils should consider the potential benefits of irrigation to reduce crop stress and increase economic yields. Currently, irrigation is not widely adopted among farmers in the region, with the irrigation area covering only approximately 6600 hectares, relying primarily on surface water (from ponds) and groundwater sources [23].

### 2.6. Estimated N Stress, Uptake, and Nitrate Leaching

Determining the optimal N rate for barley to maximize crop productivity while minimizing environmental risk requires an understanding of barley’s N use mechanisms. Therefore, two contrasting N treatments were analyzed in this study: one without fertilization (N0) and one with a locally assumed full fertilization of 100 kg ha^−1^ (N100) applied nitrogen. Consequently, the simulated N availability index, N uptake, and nitrate leaching out of the soil profile over the three contrasting growing seasons were analyzed and are presented in Figure 4. The average N availability index, which ranged from 1 to 0, was used to identify periods of barley N stress. As for the water stress, values of 1 indicate no N stress, while smaller values indicate increasing N stress. Unlike the analysis of water stress, the estimated N stress differed significantly between the N0 and N100 treatments as expected. In 2020, the average N stress for the entire barley growing season was 0.65 for N0, indicating mild N stress and 0.97 for N100, showing almost no N limitation under N100 fertilization. In 2021 and 2022, N stress in the N0 plots was higher than in the first year with moderate N stress of 0.54 and 0.51, respectively, while N100 also showed some mild N stress ranging from 0.79 to 0.83. When analyzing specific barley growth phases, no N stress was observed during the vegetative period for N100, except in 2022, when it reached 0.92. During the reproductive period, N stress for N100 was present in all years with 0.94 in 2020, 0.47 in 2021, and 0.65 in 2022, indicating that the second year was that with the largest N stress. In contrast, the N0 treatment showed N stress throughout the entire season, being slightly lower during the vegetative period (0.82 in 2020, 0.72 in 2021, and 0.61 in 2022) and severe during the reproductive period (0.42 in 2020, 0.26 in 2021, and 0.32 in 2022). Again, the second year was the growing season with the largest N stress during the reproductive stage.

The ability of the AgroC model to reproduce the barley N uptake by the grain, stem, and leaf is shown in Figure 4. As illustrated, simulated barley N uptake varied across years and fertilization conditions. During the 2020–2022 period, at barley physiological maturity, observed N uptake in the grain ranged from 51.2 to 58.9 kg ha^−1^ N for the N100 treatment and was significantly less for the N0 treatment, ranging from 12.8 to 18.9 kg ha^−1^ N (25 and 32% of N100 treatment). At its peak, N uptake by leaves varied between 13.5 and 20.4 kg ha^−1^ N for the N100 treatment and was approximately half that amount for the N0 treatment, ranging from 5.1 to 7.4 kg ha^−1^ N (38 and 36% of the N100 treatment). N uptake by stems in 2020–2022 ranged from 9.1 to 14.1 kg ha^−1^ N for the N100 treatment and from 2.1 to 4.2 kg ha^−1^ N for the N0 treatment (23 and 30% of the N100 treatment). In nearly all cases, the AgroC model successfully represented the observed N uptake data. However, the model underestimated N uptake by grains for both treatments in the last year. This discrepancy was likely due to the varying timing of water and N stress during the barley growing seasons (2020–2022), as well as instances when both stressors affected the barley simultaneously.

Differences in the simulated mass of nitrate leached below a 120 cm soil depth were primarily driven by year-to-year climatic variability, as shown in Figure 3, initial N status, and differences in fertilizer treatments. Simulated nitrate leaching was consistently higher in the N100 treatment than in the N0 one across all growing seasons and years. The highest nitrate leaching generally occurred during the winter months, when downward water flow occurs due to precipitation exceeding evaporation and a lack of N uptake by plants. For example, in the N100 treatment, the nitrate concentration in leachate during the 2020–2021 and 2021–2022 winter periods reached nearly 30 mg L^−1^, while in the N0 treatment, the nitrate concentration in leachate peaked at approximately 20 mg L^−1^ during the 2020–2021 winter but decreased to 15 mg L^−1^ in the following winter. This higher nitrate concentration in leachate during the first winter season was likely influenced by the relatively high initial soil inorganic N reserves before barley sowing, which were 71.3 kg N ha^−1^ (NO3-N + NH4-N) even though the model has been spun up over 5 years to avoid such artifacts. Leaching during the autumn period was also substantial. For example, in the N100 treatment, the N concentration in leachate in 2020–2021 ranged from 9.4 to 21.7 mg L^−1^, while in the following autumn season, it decreased to 9.3–13.4 mg L^−1^. In comparison, the N0 treatment showed lower values, ranging from 7.0 to 16.2 mg L^−1^ in 2020–2021 and 6.7 to 9.3 mg L^−1^ in the next autumn.

During the summer, nitrate leachate concentrations were lower, ranging from 2.4 to 10.0 mg L^−1^ for both treatments across all years. In spring, the simulated nitrate concentration in leachate was generally low, except in 2021, when it reached 30.0 mg L^−1^ for the N100 treatment and 20.7 mg L^−1^ for the N0 treatment. This higher nitrate leaching in spring 2021 was likely due to the unfavorable environmental conditions in 2020, which resulted in barley not fully utilizing the applied nitrogen. Consequently, in spring, with the soil not covered by any vegetation, much of the remaining nitrate was leached downwards.

### 2.7. Model Application to Assess the Environmental Impact and Effect of Increasing Fertilizer on Barley Yields

#### 2.7.1. Yield Potential, Water-Limited Yield, Nitrogen-Limited Yield, N Uptake, and N Leaching

The predicted potential barley yield (without water and N stress) was (in t ha^−1^) 4.80 in 2020, 6.02 in 2021, and 4.94 in 2022 (Figure 5). These potential yield fluctuations were mainly influenced by the different meteorological conditions between the growing seasons and associated changes in solar radiation and temperature.

In the next step, the simulated yield for the N100 treatment over three growing seasons was compared to potential yield without water stress. The grain yield losses due to water shortage were significant in 2020, reaching 2.61 t ha^−1^ (−54.4%). In 2021, the yield losses due to water stress were only 0.65 t ha^−1^ (−10.8%), indicating more favorable meteorological conditions compared to the other years. In 2022, the yield losses amounted to 2.23 t ha^−1^ (−45.1%), which was slightly smaller than those estimated for the first year.

Nitrogen stress reduced the grain yield significantly by 2.60 t ha^−1^ (−54.3%) in 2020, by 2.92 t ha^−1^ (−59.2%) in 2021, and by 3.00 t ha^−1^ (−49.9%) in 2022, whereby year to year fluctuations were less pronounced compared to water stress impacts. The calculated barley yield gap on the marginal sandy soil, influenced by the combined effects of water and N stress, was 56.9% in 2020, 50.5% in 2021, and 67.4% in 2022. This indicated that even under favorable environmental conditions (second year with water yield losses of only −10.8%), the grain yield reduction due to the combined stresses was large.

The potential yield and yield gap results provided key insights into the interaction between environmental conditions, N management, and water availability, which are all crucial for optimizing barley production on sandy soils. This study evidenced potential barley grain yields ranging from 4.8 to 6.02 t ha^−1^ (dry weight) on sandy soil at a latitude of 54° N in the Baltic region. According to the Global Yield Gap Atlas (www.yieldgap.org), the estimated barley yield potential in Lithuania ranged from 7.0 to 7.2 t ha^−1^, depending on climate zones. However, the Atlas information does not account for soil types, making direct comparisons with our results less applicable. Despite barley’s inherent tolerance to abiotic stress [28], our findings showed that N and water stress during growth largely reduced yields on sandy soils across all experimental years. Yield losses were even more severe in unfertilized plots, with losses of 81.2, 83.3, and 81.7% in 2020, 2021, and 2022, respectively, under full N (N0) and water stress. According to the Global Yield Gap Atlas, average barley yield losses in Lithuania are around 49.2%. Across Europe, under highly contrasting climate conditions, rainfed barley absolute yield gaps vary between 1.0 and 7.3 t ha^−1^. This corresponds to a relative yield gap ranging from 12% to 75% of the potential yield [29]. In general, yield gaps can vary between years [30] as also shown in our study and depend on N and water management [31].

In a next step, the optimal N fertilization rates in which N does not limit the barley yield potential were analyzed using the results from the synthetic modeling study, which showed that the N fertilization rates were increased in steps of 10 kg N ha^−1^ from no fertilization (N0) to 200 kg N ha^−1^ (N200). Here, the fertilization was always split into two applications as done for the field experiments. Again, a five-year spin-up prior to the year 2020 was added to account for changes in soil N status due to the different fertilization rates. In Figure 5, the grain yield at harvest, N uptake for the different fertilization levels, as well as the N leaching for the N0, N100, and N200 scenario and individual years are depicted. In 2020, the N fertilization rate for optimal harvest was 110 kg ha^−1^, which was a slightly higher fertilization amount than used in the field experiments, which was assumed to be optimal for this site. However, in 2021 and 2022 the optimal rate increased to 170 kg ha^−1^, indicating that higher N inputs were needed under more favourable climatic conditions for optimal barley production. Irrespectively of those high optimal N fertilization rates obtained from the simulation results, farmers should consider the economic and environmental optimal N rate, which are often lower than the agronomic maximum [32]. The simulated barley N uptake by grains showed also variation among the years, likely due to differences in simulated grain yields (Figure 5). In 2020, N uptake under the N0 treatment was 15.9 kg N ha^−1^, increasing to 54.79 kg N ha^−1^ with higher N fertilization (up to N110), where it reached saturation. Conversely, in 2021, likely due to favourable soil moisture conditions, N uptake increased from 23.9 to 126 kg N ha^−1^ for the N0 to the N180 rate. In 2022, N uptake ranged from 9 to 65.4 kg N ha^−1^ for the same rates. Accordingly, the N fertilization limit was N110 in 2020, while it was much higher in 2021 and 2022, reaching N170. This outcome is highly relevant to Lithuania and other Baltic–Nordic countries, where the political focus is currently on reducing N application to protect surface and groundwater bodies as requested by EU policies [33]. Indeed, farmers often use ‘excessive’ fertilization as a form of insurance to generate optimal yields, but overfertilization can result from overestimating expected yields or neglecting local soil conditions, which are typically the basis for N fertilizer recommendations. For example, a meta-analysis of N fertilizer experiments in Finland showed that current fertilization recommendations for spring cereals, including barley, which are based on growers’ yield expectations, can lead to significant errors in N management practices [34]. According to Lithuania’s national fertilization recommendations, N fertilizer applications to achieve a barley grain yield of 1 t ha^−1^ are 21 kg of N, 10 kg of P, and 21 kg of K [35]. However, our experiments showed that barley’s N requirements were slightly higher, at 23 to 24 kg of N per ton of grain yield. Additionally, these national recommendations are based on a standard average barley yield of 4.4 t ha^−1^, which are higher than those recorded over this 3-year study years on the marginal soil. Looking at the negative site effects of fertilization in terms of N leaching to groundwater, one can see that the barley that received the highest rate of N fertilization (200 kg ha^−1^) exhibited the highest NO3- leaching rate over three growing seasons (Figure 5). As shown, during January to March 2021, nitrate leaching fluctuated between 48 and 56 mg N L^−1^, exceeding the maximum allowable limit of 50 mg N L^−1^ [36]. In contrast, nitrate leaching for N100 was approximately half of that, around 30 mg N L^−1^, while for N0, it was even lower at 20 mg N L^−1^. In February 2022, nitrate leaching for N200 remained elevated, ranging from 34 to 44 mg N L^−1^, and again exceeded the allowable limit in March, reaching 56 mg N L^−1^. The risk of nitrate leaching is closely linked to the excessive use of fertilizers, particularly occurring after the growing season and within the crop cycle [37].

#### 2.7.2. Estimated Nitrogen Stress and Nitrogen Use Efficiencies

The average N stress intensity over the entire barley growing season along with various indicators reflecting the N use efficiency are plotted for the different N fertilization rates in Figure 6. As shown, N stress intensity was influenced by both fertilization rates and yearly environmental conditions and reflected exactly what has been discussed for the grain yield and N taken up by the biomass plotted in Figure 5. In general, the least N stress occurred in the year 2020 for N0, and low fertilization rates were observed, as the demand (uptake) by the plant was low, reflecting low productivity (grain yield). In contrast, in years with more favorable conditions, N stress increased slightly for low fertilization levels but all levelled out at a certain threshold (optimal fertilization) level.

The calculated agronomic nitrogen efficiency coefficient (AE_N_) is useful when investigating the economic portion of plants (grain yield). The overall trend of the AE_N_ coefficient showed that 2021 had the highest values, indicating more favorable growing conditions compared to 2020 and 2022. Additionally, at higher N fertilization rates, AE_N_ values increased up to a certain point before reaching a peak and then decreasing again. In 2020, AE_N_ increased from 4.7 at a low fertilization of N10 to 14.3 at N100. Beyond N100, N efficiency started to decline, suggesting that additional N beyond N100 did not translate to a proportional yield increase. In contrast, in 2021, AE_N_ significantly increased from 7.4 at a low fertilization of N10 to 24.4 at N180, indicating a strong response of yield to N fertilization. In 2022, the calculated AE_N_ coefficient increased from 4.1 at N10 to 13.1 at N170, followed by a gradual decline beyond this rate.

The partial factor productivity of applied nitrogen (PFP_N_) provides information on the ratio between produced barley grain yield and applied N. Therefore, it can indicate the crop’s capability to convert N inputs into outputs (yield). The overall trends shown in Figure 6 indicate that the highest PFP_N_ values were at the N10 rate across all years, suggesting low N rates were more efficient in terms of productivity per unit of applied N compared to higher rates. Additionally, all years showed almost the same pattern of PFP_N_, whereby for the year 2021, the low slope at medium and higher fertilization rates showed a higher PFP_N_ compared to the years 2020 and 2022. This means that for the year 2021, more favorable climatic growth conditions led to a better N use efficiency, as indicated by the exceptionally high values compared to the other years. The low slope of PFP_N_ at medium to higher fertilization rates suggests a diminishing return on N use efficiency at higher fertilization rates. Finally, the crop recovery efficiency of applied nitrogen (RE_N_) showed the proportion of applied N that was taken up and utilized by the crop. In general, the RE_N_ trends were very similar to those observed for AE_N_, showing that in all three years, RE_N_ values increased with higher N rates up to a peak and then began to decrease beyond at higher fertilization rates (see Figure 6). Similar to the AE_N_ and PFP_N_ coefficients, RE_N_ indicated that the most favorable year for barley cultivation was 2021.

The highest RE_N_ values in 2020 were calculated for the N110 rate, with RE_N_ reaching 0.53. In 2021, RE_N_ peaked at a much higher fertilization rate of N170 with a RE_N_ of 0.88, while in 2022, it peaked at a fertilization rate of N170 with 0.49. However, all calculated N use efficiency indices did not account for the initial N reserves in the soil, which might confound the different years, even though care was taken to be independent on initial N stocks in the soil by adding a 5-year spin-up period in the simulation.

#### 2.7.3. Estimated Water Stress and Water Use Efficiency

In order to evaluate the agronomic water use efficiency (WUE) for barley grown on the marginal sandy soil, the average water stress intensity for the entire barley growing season along with WUE was calculated and plotted in Figure 7 for the different N fertilization rates. In contrast to the N stress described above, the water stress intensity remained almost constant over the entire range of N fertilization rates and only differed slightly among the three years. For example, in 2020, water stress ranged from 0.75 to 0.77; in 2021, it was slightly lower with values ranging from 0.84 to 0.86; and in 2022, it showed the lowest values among all years at 0.73–0.75 over all fertilization rates. Water stress and its intensity mainly indicate the feedback between available water in the soil system replenished by precipitation and water losses by the plant due to transpiration. Classically, transpiration is coupled to photosynthetic activity or carbon dioxide assimilation for biomass growth as both carbon dioxide and water vapor are exchanged through the leaf stomata. Therefore, actual ET is an important indicator of crop water requirements and use [38]. However, for the sandy soils in the present study, a severe amount of water stress was given, no matter how much N fertilization was applied (see top panel, Figure 3).

The WUE values generally increased with higher N fertilization rates, then reached a peak, which corresponded to the maximum yield reached for the higher fertilization amounts and remained stable across all years (Figure 7). The highest WUE values (peaks) varied each year, reflecting the influence of yearly environmental conditions and the maximum yield reached in the individual years. In 2020, with the lowest total precipitation, WUE values for the N0 treatment were 2.88 kg ha−^1^ mm−^1^, reaching a maximum of 9.72 kg ha−^1^ mm−^1^ at the N120 fertilization rate. In 2021, with increased precipitation, WUE values were higher, ranging from 2.98 to 16.40 kg ha^−1^ mm^−1^. Despite the highest precipitation in 2022, WUE values were not the highest, ranging from 1.01 to 7.82 kg ha^−1^ mm^−1^. Accordingly, the total sum of precipitation over the growing season did not solely determine the WUE rather than the distribution over the season. In 2022, eight high rainfall events of 10.3, 11.3, 13.9, 14.5, 29.3, 30.8, 35.5, and 50.4 mm contributed to a total of 196 mm of precipitation. Sandy soils, due to their large pores and high hydraulic conductivity at larger water contents, are known to exhibit also low water retention, allowing water to infiltrate and drain quickly through the soil profile. Consequently, the infiltrated water may not be available for plant growth. As a result, a significant portion of the 2022 precipitation would be ineffective in maintaining barley growth and yield formation.

Although in many cases, water stress can also lead to N stress, both stressors can occur simultaneously under specific conditions, such as suboptimal water supply and insufficient fertilization. Therefore, the relationships among WUE, N stress intensity, and N uptake were analyzed by correlating WUE with the N stress intensity or plant N uptake (see Figure 7). In all 3 years, a strong positive correlation was found between simulated WUE and simulated N stress intensity (R^2^ = 0.97–0.99), as well as between simulated WUE and simulated N uptake into grains (R^2^ = 0.99). Higher WUE values were associated with higher N availability, indicating that WUE increased when barley experienced less N stress during growth. A similar trend was observed between WUE and N uptake, emphasizing the interdependence of water and N efficiency in barley cultivation.

## 3. Materials and Methods

### 3.1. Site Description

According to the environmental stratification of Europe, Lithuania falls within the Nemoral climate zone. This climate zone is characterized by a continental and rather cool climate, as well as a relatively short vegetation period. The same climate zone encompasses the southern portion of Scandinavia, the Baltic states, and Belarus [39]. Lithuania’s landscape territory has notable variations in terms of air temperature, precipitation pattern, and soil types. The mean annual air temperature ranges from 5.8 to 7.6 °C, while annual precipitation spans from 550 to 910 mm [40]. The predominant soil classification encompasses Luvisols, Cambisols, Gleysols, and Arenosols, constituting 28.5, 15.9, 14.6, and 13.2% of the land area, respectively.

The locations of the field trials fall into the agro-climatic zone IIIB of southeastern Lithuania [41], which has a pronounced seasonal variation with distinct spring, summer, autumn, and winter periods. This agro-climatic zone is relatively warm with moderate annual precipitation, but it is dominated by infertile (marginal) soils. The average annual air temperature in this zone is 7.2 °C and the total annual precipitation is 678 mm (mean value over the period 1991–2020). The main soil is Endocalcaric Eutric Brunic Arenosol (Geoabruptic, Aric, see Appendix A), which is predominant in Lithuania [42]. The soil texture in the top soil horizon is sandy silt loam with an average soil organic carbon content of 1.34% (see Table 2). The main soil agrochemical characteristics were a slightly acidic pH, with high amounts of plant-available phosphorus (P_2_O_5_) and potassium (K_2_O) levels (Table 2).

### 3.2. Experimental Design

The field experiment was carried out at the Lithuanian Research Center for Agriculture and Forestry in the fields of the regional Vokė Branch (54.588029° N; 25.135752° W) in a 3 year period (2020–2022). The spring barley variety KWS Fantex was sown in late April/early May at a density of 450 m^−2^ at 3–5 cm depth. The barley cultivar selected and used in the experiment was bred in Germany by the KWS Lochow GmbH Seed Company located in Bergen. It was chosen because of its high yield potential, high resistance to lodging, and very good resistance to net blotch, spot blotch, scald, and mildew, as well as double resistance to nematodes. Barley was annually harvested after the crops reached physiological maturity. The experimental design for the three years included various fertilization treatments. However, in this study, we focused on two treatments, (1) no fertilization (N0) and (2) N_100_P_80_K_140_ (N100) plots with mineral fertilization, as these treatments provided the most contrasting nutrient conditions for this field experiment. Treatments were arranged in 4 randomized blocks with a plot size of 30 m^2^, in which only 13.4 m^2^ (1.68 × 8 m^2^) of the inner parts were harvested. The mineral fertilizers were in the form of ammonium nitrate (34.4-0-0), granular superphosphate (0-20-0), and potassium chloride (0-0-60). The full dose of P and K fertilizers was applied manually before sowing, while N application was split, i.e., N_50_ was used before sowing and the N_50_ rest was applied during the tillering growth stage of barley.

### 3.3. Plant Measurements

During the barley growth period, the development stages from emergence to maturity were assessed weekly using the BBCH scale [43]. The scale’s first digit indicates the principal growth stages (e.g., 1 for leaf development and 2 for tillering), while the second digit provides details on a specific development step. A development stage for barley was designated when 50% or more of the plants across the plot had reached that particular stage.

From 2020 to 2022, the total aboveground biomass (TAB) of barley was periodically sampled throughout the growing season. Specifically, 6 TAB measurements were taken in 2020 at stages BBCH13, BBCH31, BBCH61, BBCH73, BBCH85, and at harvest. In 2021, 5 TAB measurements were conducted at BBCH21, BBCH32−37, BBCH75, BBCH85, and at harvest. In 2022, four TAB measurements occurred at BBCH23−25, BBCH47−59, BBCH75, and at harvest. For these measurements, samples were collected from each plot from a ground surface area of 0.5 m × 0.5 m (0.25 m^2^). The harvested plants were divided into their principal components (leaves, stems, and grains) as applicable on the date of sampling. Subsequently, the samples were dried in an oven at a temperature of 65 ± 5 °C until a constant weight was achieved to determine the dry biomass weights.

At barley physiological maturity (BBCH 92−99), the inner parts of experimental plots (1.68 m × 8 m = 13.4 m^2^) were harvested. Similar to the seasonal TAB measurements, at harvest, the plants were divided into their principal components and subjected to the same drying procedures. Barley leaves, stems, and grain subsamples taken at harvest were used to determine the N concentration in plant parts using the Kjeldahl method [44]. For each measured plot (N0 and N100) accumulated N uptake (based on dry matter) was calculated separately by multiplying the resulting nitrogen concentration (N%) by the corresponding yield by:(1)$$N\;uptake\;\left(kg\;{ha}^{-1}\right)=\frac{N\%\times dry\;matter\;(kg\;{ha}^{-1})}{100}$$

Additionally, three N use efficiency (NUE) coefficients were calculated: the agronomic N efficiency coefficient (AE_N_), the partial factor productivity of applied N (PFP_N_), and the crop recovery efficiency of applied N (RE_N_). These were calculated using the following equations, as described by Dobermann [45]:(2)$$AE_{N}=\frac{\left(Y_{N}-Y_{o}\right)}{F_{N}}$$(3)$${PFP_{N}=Y}_{N}-F_{N}$$(4)$$REN=\frac{\left(U_{N}-U_{0}\right)}{F_{N}}$$
where *Y_N_* is the barley yield under the applied N rate (kg DM ha^−1^), *Y*_0_ is the barley yield (kg DM ha^−1^) in a treatment with no N application, *F_N_* is the amount of N applied (kg N ha^−1^), *U_N_* is the total barley N uptake in grain yield at maturity in a treatment with applied N fertilization (kg N ha^−1^), and *U*_0_ is the total barley N uptake in grain yield at maturity in a treatment without N fertilization (kg N ha^−1^).

Leaf area was measured periodically using a simple HP printer scanner. Therefore, the plants were harvested from 0.25 m^2^ area and separated into leaves, stem, and grains. In the next step, the leaves were scanned. The scanned images of the leaves were analyzed with the WinFOLIA (Regent Instruments Canada Inc., Québec City, QC, Canada) image analysis tool to calculate the leaf area index (LAI). Overall, four LAI measurements were conducted in 2020, and three each in 2021 and 2022.

### 3.4. Soil Measurements

#### 3.4.1. Soil Chemical Properties

Before the start of each barley growing season, the nutrient status of the soil was evaluated. To conduct the soil analysis, composite samples were collected from the plowed layer (0–30 cm depth) at 12 distinct locations within the field. The chemical analyses included measurements of soil pH, soil organic carbon (SOC), total nitrogen (N_total_), plant-available phosphorus (P_2_O_5_), potassium (K_2_O), and bulk soil mineral nitrogen (SMN) as nitrate and ammonium. Additionally, two samples were taken from each of the five soil horizons for chemical analysis (Table 3). The pH of the soil samples was determined using an XS Instruments pH meter (Via della Meccanica, Italy) in a 1 M KCl solution with a 1:5 (vol/vol) ratio. The SOC content was measured through dry combustion using a Liqui TOC II instrument (Elementar, Langenselbold, Germany), with soil samples first being treated with HCl to eliminate the inorganic carbon fraction. These measurements were performed in triplicate for each sample. N_total_ was assessed using the Kjeldahl method with a semi-automatic Velp Scientifica™ UDK 139 instrument (VELP Scientifica, Usmate Velate, Italy), followed by manual titration of the sample with a 0.1 M NaOH solution. The soil P_2_O_5_ content was determined via a Shimadzu UV 1800 spectrophotometer, and the K_2_O content was measured using flame emission spectroscopy with a JENWAY PFP7 flame photometer (Thermo Scientific, Cambridge, UK) at a 766 nm wavelength. Both nutrients were analyzed using the Egner–Riehm–Domingo (A-L) method. SMN was measured (ISO 14256-2:2005) using a spectrometric analyzer (Fiastar 5000, Foss Analytical AB, Höganäs, Sweden. All chemical analyses of the soil were performed at the Agrochemical Research Laboratory of the Lithuanian Research Centre for Agriculture and Forestry.

#### 3.4.2. Soil Hydraulic Properties

In 2020, before the start of the barley growing season, a pit was manually dug in the central experimental location, and the soil was classified as an Endocalcaric Eutric Brunic Arenosol (Geoabruptic, Aric) with five main horizons according to WRB [42]. A photo of the profile is provided in Appendix A. For soil physical characterization, 10 undisturbed soil samples of 250 cm^3^ were taken as duplicates from the horizons at 15–20, 40–45, 60–65, 90–95, and 110–115 cm. The soil hydraulic properties were determined by the evaporation method using the HYPROP^®^ system (Meter Group, München, Germany) [46] in combination with the WP4^®^ Dewpoint Potentiometer (Decagon Devices, WA, USA). Saturated hydraulic conductivity, K_s_, was measured using a falling head using the KSAT system (Meter Group, München, Germany). The estimated van Genuchten [47] soil hydraulic parameters for the soil profile up to a depth of 120 cm are listed in Table 4.

Additionally, at all horizons, samples were taken to determine the soil texture and bulk density using the same undisturbed samples as used for the evaporation method (Table 5). Soil texture was analyzed according to DIN ISO 11277 [48] by wet sieving and the pipette method.

#### 3.4.3. Soil Water Content

From 2020 to 2022, the soil volumetric water content (SWC) was monitored hourly at 5 different depths (15, 40, 60, 90, and 110 cm). For the SWC measurements, a wireless sensor network, SoilNet [49], was installed prior to seed bed preparation. In the AgroC model, daily averaged SWC values, aggregated from the hourly raw data, were used.

Barley water use efficiency (WUE) was calculated using the equation described by Howell [50]:(5)$$WUE=\frac{Barley\;grain\;yield}{Water\;used\;to\;produce\;the\;yield}$$

### 3.5. AgroC Model Setup and Calibration

For the simulation of barley growth, the AgroC model [21,22] was used. The AgroC model is a one-dimensional state-of-the-art agroecosystem model, which has been applied for different crops, such as winter wheat [22], maize [31,51], sugar beet [52], oat, barley [53], and hemp [20]. For the water flow, it solves the Richards equation, and the Mualem van Genuchten functions [47] were used for the soil hydraulic parameterization. More model details can be found in Herbst et al. [21] and Klosterhalfen et al. [22]. The detailed description of the AgroC N module is provided in Appendix B.

The AgroC model was set up to mimic the field experiments, in which the simulation domain was set to 150 cm depth and the 5 pedogenetic horizons, as listed in Table 3, were included. The upper boundary was defined by the atmospheric conditions with precipitation and given potential evapotranspiration (ET_pot_). The lower boundary was set to free drainage. Crop management (seeding, seeding density, harvest, and N-fertilization) followed the experimental data.

Model calibration was performed in two steps. First, phenology was calibrated using the approach suggested by Wallach et al. [54], in which the plant parameters, the input used [55] for barley simulation in western Germany, were taken as start parameters. Secondly, the soil hydraulics were optimized for each pedogenetic soil layer identified in the field using the measured soil water contents to be minimized in the objective function. To optimize the phenology and the soil hydraulic parameters, the Shuffled Complex Evolution (SCE-UA) algorithm [56,57] was used. As the soil hydraulic parameters were measured in the laboratory those parameters were used and only the saturated hydraulic conductivity K_s_ was inversely estimated for each horizon, which minimized the number of parameters to assess. Please note that only one set of soil hydraulic parameters for the three plot locations, where the experiments were performed, has been optimized, as the small-scale variability in this area is typically low. Finally, the crop parameters were optimized manually to match observed biomasses, yields, and LAIs.

For all calibration steps, the first and third years (2020 and 2022) were selected, as those years are most contrasting in climate, fulfilling the requirements for calibration as suggested by Wallach et al. [58]. The remaining second year (2021) was selected for model validation. For the phenology and biomass (yield) calibration, all fertilization trial (N100) data were used. To calibrate the nitrogen uptake and distribution within the plant organs, both the N100 and no fertilization data for the N0 trials were used. As the soil nitrate and ammonia concentrations were initialized by measured data, which had to be interpolated and are known to be uncertain, the model was spun up for a 5-year period assuming the same management as for the experimental years (same crop, same seeding and harvest dates, and same fertilization). For the spin-up, the measured climatic data of the three years were looped. An overview of the calibration steps and calibrated parameters is provided in Appendix A.

### 3.6. Climate Data

The weather data for this study were obtained from the Vilnius meteorological station, located 5.3 km away from the barley experimental site. The daily data utilized for simulations included precipitation (mm), maximum and minimum air temperature (T_max_ and T_min_) (°C), relative humidity (%), wind speed at 2 m height (m s^−1^), and solar radiation (J cm^2^ day^−1^). Using this data, the reference Penman–Monteith evapotranspiration (ET_0_) (mm day^−1^) was calculated [59]. These weather data were sourced from the Lithuanian Hydrometeorological Service, through its data sharing service system (available at https://www.meteo.lt, accessed on 8 January 2024).

### 3.7. AgroC Model Application

#### 3.7.1. Prediction of Barley Yield Potential and Yield Gap

The impact assessment of the different environmental conditions on barley yield was performed sequentially through simulations for the years 2020–2022. In a first step, the potential yield was estimated by disabling the plant’s response mechanisms to nitrogen and water, thereby removing abiotic stress factors and assuming ideal fertilization and watering conditions. In a second step, the impact of water stress was simulated, in which the nitrogen response was deactivated, isolating the effect of water only. In the next step, the N impact on yields was assessed, by disabling the water stress and focusing solely on the N impact. In the final step, actual barley yields were simulated by accounting for N and water stress. The comparison of potential yields with those derived from the latter three steps provided insight into the yield reductions caused by N stress, water stress, and the combination of both stressors.

#### 3.7.2. Evaluation of N Management Practice Scenarios

To evaluate the effect of increasing N fertilization rates from N0 (no fertilization) to N200 (200 kg N ha^−1^) and climatic variables on barley total aboveground biomass, grain yield, water and N stress intensity, N uptake, and N leaching, the previously calibrated and validated AgroC model was used for simulations. The N fertilization scenarios were simulated for the same years in which the model was parameterized. For the different assumed N application levels of 10 to 200 kg ha^−1^ per growing seasons, 19 simulations were performed with N application steps of 10 kg ha^−1^ each (10, 20, 30……190, and 200 kg ha^−1^). For fertilization, the same application dates as for the field experiments were used, and the assumed total N amounts were split in half for the first and second applications.

### 3.8. Statistical Analysis

The prediction capability of the model for both calibration and validation was tested by the Willmott index of agreement (*d*), root mean squared error (RMSE), and BIAS, computed as follows [60]:(6)$$d=1-\frac{\sum_{i=1}^{n}{\left(Y_{i}-X_{i}\right)}^{2}}{\sum_{i=1}^{n}{\left(\left|Y_{i}-M0\right|+\left|X_{i}-M0\right|\right)}^{2}}$$(7)$$RMSE=\sqrt{\frac{1}{n}\sum_{i=1}^{n}{\left(y_{i}-x_{i}\right)}^{2}}$$(8)$$BIAS=1/n\sum_{i=0}^{n}\left(y_{i}-x_{i}\right)$$
where *Y_i_* is the simulated value; *X_i_* is the measured value; *MO* is the average value of “*n*” measured values; and *n* is the number of measurements.

The index of agreement (*d*) developed by Willmott [60] as a standardized measure of the degree of model prediction error varies between 0 and 1. A value of 1 indicates a perfect match between observation and the model and 0 indicates no agreement at all. The BIAS measures the average difference between measured and simulated values. A positive BIAS indicates model under-prediction, and a negative BIAS indicates over-prediction. A lower RMSE indicates a better fit to the observed data compared to larger values.

## 4. Conclusions

In this study, the AgroC model was calibrated and validated using a multiyear barley field experiment on marginal sandy soil under varying fertilization and weather conditions. After calibration, the model accurately simulated barley development, grain yield, and N uptake. It was then used to predict N leaching, assess N and water stress effects on yields, and analyze potential yield and yield gaps beyond what experimental data alone could provide.

This study provided experimental and modeling evidence of potential barley grain yield levels of 4.8–6.02 t ha^−1^ (dry weight) on marginal soil at a latitude of 54° N in the Baltic region. However, a significant barley yield gap was identified, primarily due to limited water and nitrogen availability in these poor sandy soils. To address this, farmers should integrate fertilization and irrigation management practices. The combined effects of water and full N stress on the sandy soil reduced barley yields by more than 80% for all three experimental years pointing to the large stress pressure at this site. To explore optimal fertilization rates, a systemic study was performed based on the calibrated model, in which the modeling results for different fertilization scenarios suggested that higher N fertilization has the potential to increase actual yields, although this increase varied between the years making general recommendations complicated. The highest barley yields were achieved with N rates between 110 and 180 kg ha^−1^. However, with increasing N fertilization rates, fertilizer efficiency declined, while nitrate leaching intensified. Therefore, the optimal and economically beneficial N fertilization rate for barley on this sandy soil should be between 100 and 120 kg ha^−1^.

This study contributed to the ongoing efforts to develop effective strategies for reducing yield gaps, ensuring food security, and promoting sustainable agricultural development in the Baltic–Nordic region.

## Figures and Tables

**Figure 1 plants-14-00704-f001:**
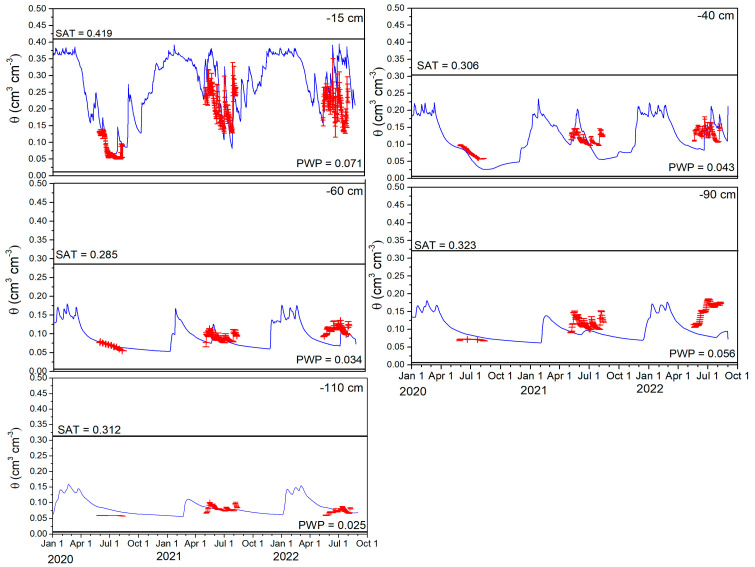
Comparison between observed (dots) and simulated (lines) volumetric soil water content, θ (cm^3^ cm^−3^) at 15, 40, 60, 90, and 110 cm depths for the N100 treatment during calibration (2020 and 2022) and validation (2021) periods for rainfed barley. SAT is the saturated water content, PWP is the water content at the permanent wilting point, and SAT-PWP is the plant available water (PAW) (all in cm^3^ cm^−3^). Error bars indicate standard deviation of the measurements.

**Figure 2 plants-14-00704-f002:**
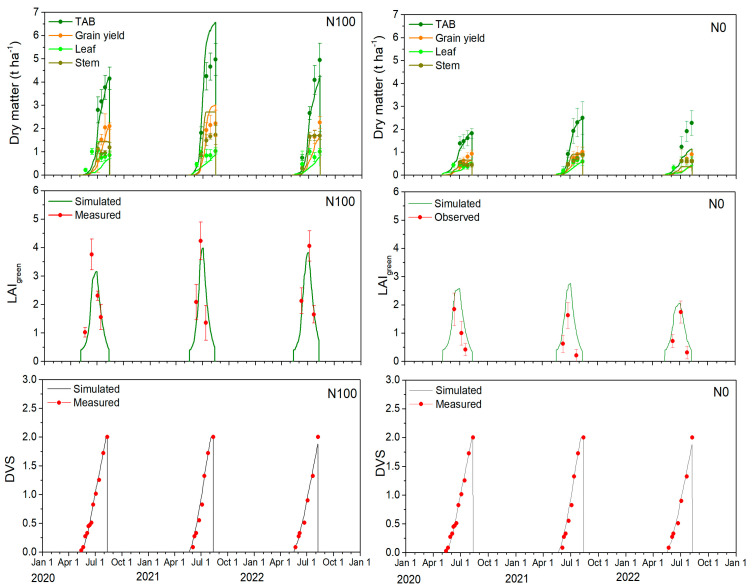
(**top**) Observed and simulated dry matter biomass of individual vegetative and reproductive parts of barley (leaves, stems, and grain yield). (**middle**) Seasonal variation in the green leaf area index (LAI) and (**bottom**) barley development stages (DVS) across 3 growing seasons, 2020–2022. In all figures, observed values are represented by dots, with error bars indicating the standard deviation, while simulated values are shown as lines for the N100 (left panel) and N0 (right panel) fertilization.

**Figure 3 plants-14-00704-f003:**
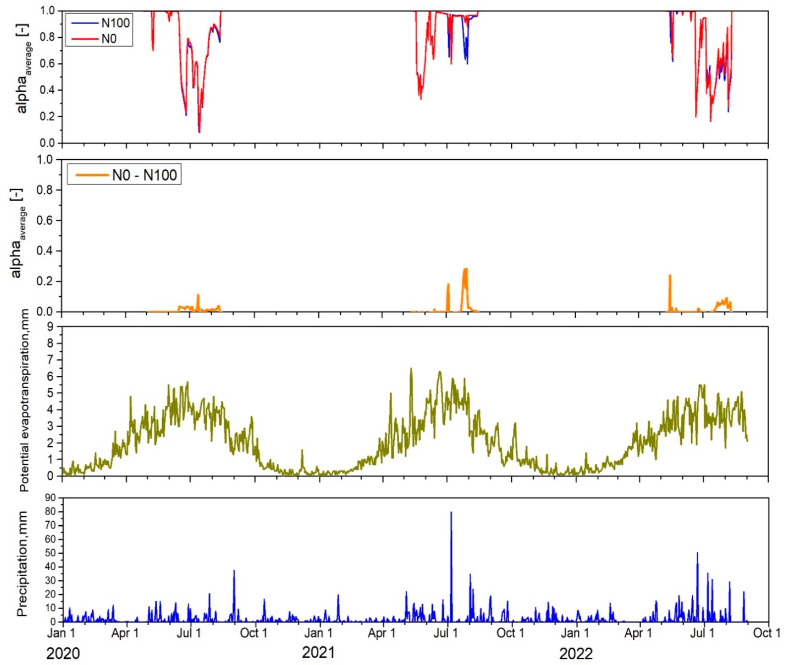
Estimated water availability index (α_avg_) for both fertilization treatments (N0 and N100), along with the difference in α_avg_ between N0 and N100, potential evapotranspiration (ET_p_), and precipitation (P) for the barley growing seasons from 2020 to 2022.

**Figure 4 plants-14-00704-f004:**
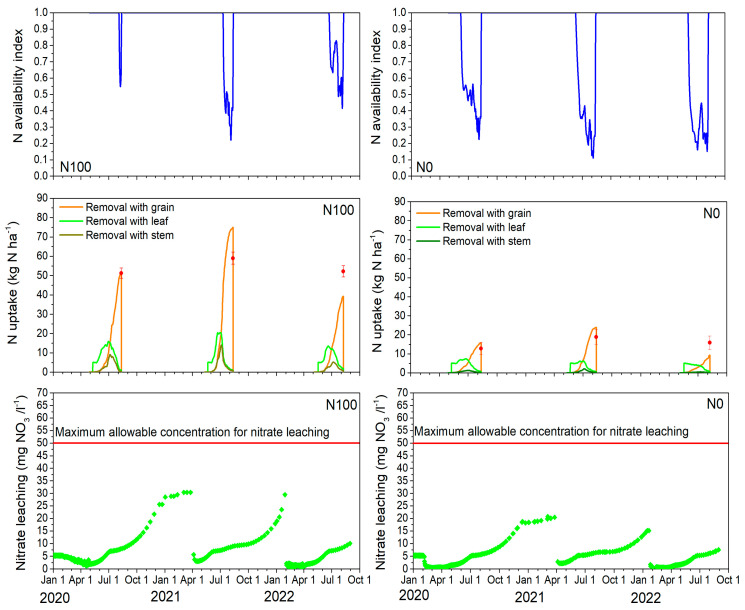
Simulated nitrogen availability index for both fertilization treatments (N100 left panel and N0 right panel), along with seasonal N uptake per organ and nitrate leaching during the barley growing seasons of 2020–2022. In the top and middle figures, observed values are represented by dots with error bars indicating the standard deviation, while simulated values are shown as lines.

**Figure 5 plants-14-00704-f005:**
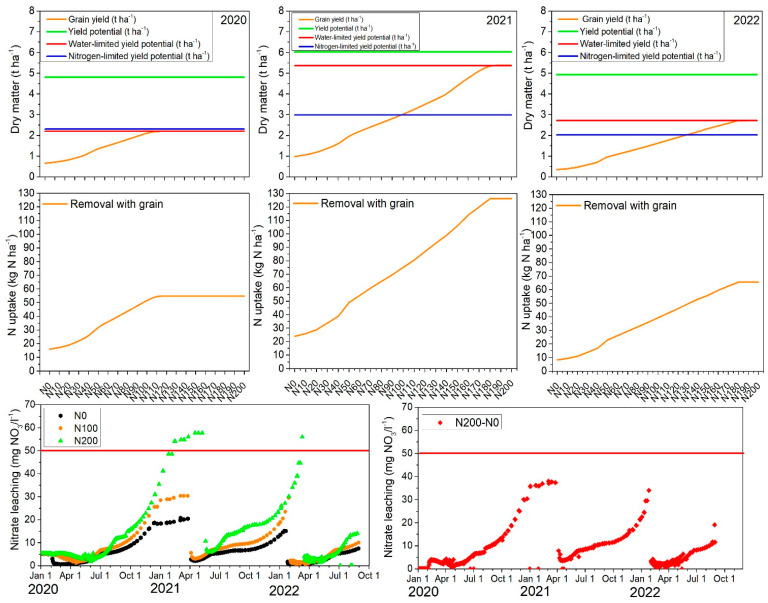
Simulated barley grain yield potential (no stress), water-limited, nitrogen-limited (**top row**), and simulated nitrogen uptake by grains (**middle row**) and simulated nitrate leaching among three contrasting treatments: N0, N100, and N200.

**Figure 6 plants-14-00704-f006:**
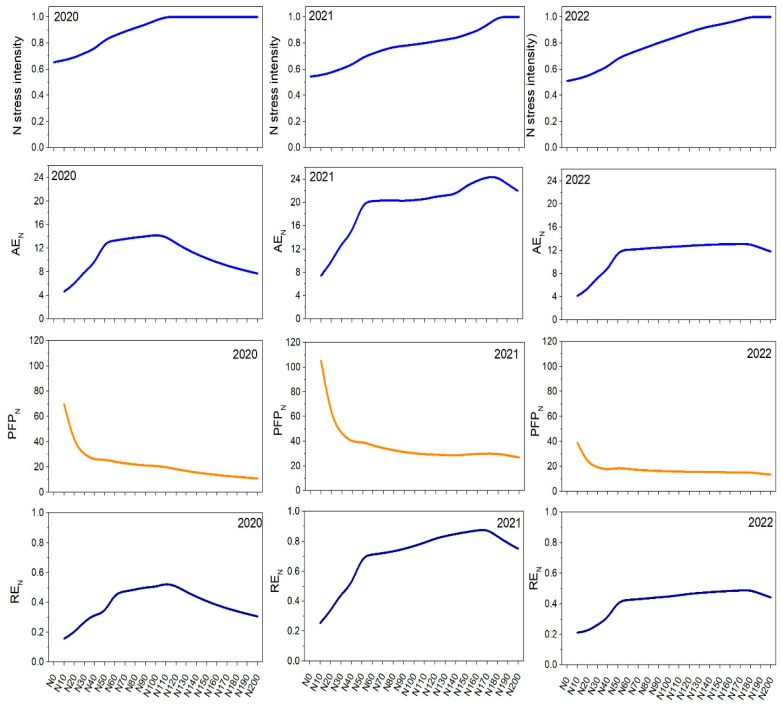
Estimated average nitrogen stress index over the growing season (**top row**) and nitrogen use efficiency, agronomic N use efficiency (AE_N_), partial factor productivity of applied N (PFP_N_), and crop recovery efficiency of applied N (RE_N_) with varying fertilization rates from 0 (N0) to 200 (N200) kg ha^−1^.

**Figure 7 plants-14-00704-f007:**
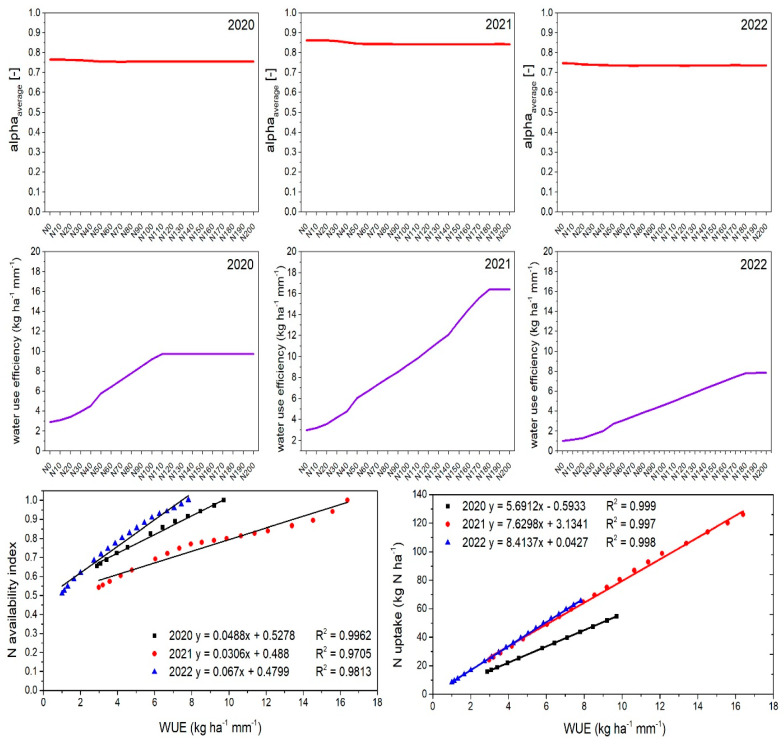
Estimated water availability index (α_avg_) (**top row**), water use efficiency (WUE) (**middle row**), and the correlation between the N availability index and WUE, with varying fertilization rates from 0 (N0) to 200 kg ha^−1^ (N200), as well as N uptake and WUE (**bottom row**).

**Table 1 plants-14-00704-t001:** Statistical values between the measured vs. simulated data for the calibration (2020 and 2022) and validation period for rainfed barley (treatments: N0 and N100) at the experimental Vokė site.

Parameters	*d*		RMSE		BIAS	
	N0	N100	N0	N100	N0	N100
2020						
Leaf area index	0.704	0.834	0.663	0.535	−0.619	0.286
TAB (t ha^−1^)	0.982	0.980	0.128	0.422	−0.029	−0.321
Leaf (t ha^−1^)	0.702	0.513	0.193	0.444	0.074	0.347
Stem (t ha^−1^)	0.892	0.587	0.172	0.446	−0.163	−0.420
Storage organs and GY (t ha^−1^)	0.725	0.928	0.241	0.315	0.258	0.242
SWC at 15 cm (cm^3^ cm^−3^)	−	0.788	−	0.030	−	0.040
SWC at 40 cm (cm^3^ cm^−3^)	−	0.775	−	0.019	−	−0.017
SWC at 60 cm (cm^3^ cm^−3^)	−	0.907	−	0.004	−	0.002
SWC at 90 cm (cm^3^ cm^−3^)	−	0.159	−	0.013	−	0.012
SWC at 110 cm (cm^3^ cm^−3^)	−	0.101	−	0.019	−	0.018
2021						
Leaf area index	0.719	0.806	0.795	0.875	−0.787	0.037
TAB (t ha^−1^)	0.998	0.925	0.051	1.197	0.034	−0.973
Leaf (t ha^−1^)	0.728	0.755	0.196	0.365	0.169	0.323
Stem (t ha^−1^)	0.802	0.570	0.159	0.938	−0.100	−0.786
Storage organs and GY (t ha^−1^)	0.975	0.643	0.166	0.624	0.027	−0.430
SWC at 15 cm (cm^3^ cm^−3^)	−	0.788	−	0.067	−	0.040
SWC at 40 cm (cm^3^ cm^−3^)	−	0.391	−	0.039	−	−0.006
SWC at 60 cm (cm^3^ cm^−3^)	−	0.468	−	0.019	−	−0.005
SWC at 90 cm (cm^3^ cm^−3^)	−	0.339	−	0.031	−	−0.025
SWC at 110 cm (cm^3^ cm^−3^)	−	0.455	−	0.008	−	−0.003
2022						
Leaf area index	0.882	0.949	0.384	0.429	−0.349	0.295
TAB (t ha^−1^)	0.629	0.972	0.772	0.527	0.656	0.485
Leaf (t ha^−1^)	0.389	0.582	0.334	0.423	0.314	0.385
Stem (t ha^−1^)	0.438	0.992	0.248	0.113	0.248	0.069
Storage organs and GY (t ha^−1^)	0.408	0.914	0.411	0.470	0.266	0.057
SWC at 15 cm (cm^3^ cm^−3^)	−	0.452	−	0.071	−	0.093
SWC at 40 cm (cm^3^ cm^−3^)	−	0.308	−	0.043	−	−0.006
SWC at 60 cm (cm^3^ cm^−3^)	−	0.293	−	0.035	−	−0.027
SWC at 90 cm (cm^3^ cm^−3^)	−	0.108	−	0.072	−	−0.065
SWC at 110 cm (cm^3^ cm^−3^)	−	0.751	−	0.011	−	0.004

Note: SWC—soil volumetric water content, *d*—index of agreement, RMSE—root mean squared error, BIAS—systematic bias index.

**Table 2 plants-14-00704-t002:** Soil properties at the experimental site and for barley management information.

	2020	2021	2022
Soil (FAO classification)	Endocalcaric Eutric Brunic Arenosol (Geoabruptic, Aric)		
Soil pH (1 N KCl extraction)	5.5	6.0	6.3
Soil P_2_O_5_ (mg kg^−1^) (Egner-Riehm-Domingo (A-L))	170	205	192
Soil K_2_O (mg kg^−1^) (Egner-Riehm-Domingo (A-L))	324	174	180
Soil organic carbon (%)	1.34	1.16	1.21
Soil N total (%) (Kjeldahl)	0.103	0.101	0.103
Previous crop	Buckwheat	Barley	Barley
Barley cultivar	KWS Fantex	KWS Fantex	KWS Fantex
Barley seeding dates	27 April 2020	10 May 2021	04 May 2022
Seeding density (seeds per m^2^)	450	450	450
Plot size	3 m × 10 m = 30 m^2^	3 m × 10 m = 30 m^2^	3 m × 10 m = 30 m^2^
Fertilization	27 April 2020—N_50_P_80_K_140_; 8 June 2020—N_50_	7 May 2021—N_50_P_80_K_140_; 11 June 2021—N_50_	3 May 2022—N_50_P_80_K_140_; 6 June—N_50_
Pesticides, fungicides, insecticides	–	–	–
Barley harvesting	12 August 2020	13 August 2021	9 August 2022

**Table 3 plants-14-00704-t003:** Soil chemical properties determined before barley planting in 2020 at different soil horizons: Soil organic carbon (SOC), pH, total nitrogen (N_total_), plant-available phosphorus (P_2_O_5_), plant-available potassium (K_2_O), soil mineral nitrogen (SMN) including nitrate nitrogen (NO_3^−^_-N), and ammonium nitrogen (NH_4^+^_-N).

Horizon Description	SOC	pH	N_total_	P_2_0_5_	K_2_0	SMN
						(NO_3^−^_-N + NH_4^+^_-N)
	%	-	%	mg kg^−1^	mg kg^−1^	kg ha^−1^
Ap (0–30 cm)	1.34	5.5	0.103	170	324	71.3 ± 11.5
B1 (30–50 cm)	0.46	5.8	0.013	120	153	–
B2 (50–78 cm)	0.54	6.4	0.014	259	122	–
2C(k) (78–105 cm)	0.37	7.6	0.003	195	43.5	–
2C (105–120 cm)	0.29	7.6	0.003	208	38	–

Horizon description: Ap = accumulative plaggic horizon in which decomposed organic material is being accumulated; B1 and B2 = illuvial horizons; 2C(k) and 2C = initial mineral horizons.

**Table 4 plants-14-00704-t004:** Soil horizons and soil hydraulic parameters measured at the barley experimental location in 2020.

Horizon Description	PWP	*θ_r_*	*θ_s_*	*α*	*n*	*K_s_*
	(cm^3^ cm^−3^)			(cm^−1^)	(-)	(cm day^−1^)
Ap (0–30 cm)	0.071	0.0145	0.419	0.011	1.395	58.87
B1 (30–50 cm)	0.043	0.0260	0.306	0.033	1.980	9.34
B2 (50–78 cm)	0.034	0.0205	0.285	0.040	3.647	5.88
2C(k) (78–105 cm)	0.056	0.0240	0.323	0.088	2.821	5.05
2C (105–120 cm)	0.026	0.0260	0.312	0.047	4.575	1.95

The soil hydraulic parameters according to van Genuchten [42] are: *θ_r_* = residual water content; *θ_s_* = saturated water content; *a* = inverse air entry pressure; *n* = shape parameter; and *K_s_* = saturated hydraulic conductivity estimated by calibration. PWP is the water content at the wilting point. Horizon description: Ap = accumulative plaggic horizon in which decomposed organic material is being accumulated; B1 and B2 = illuvial horizons; 2C(k) and 2C = initial mineral horizons.

**Table 5 plants-14-00704-t005:** Soil horizons, soil texture, and bulk density measured in 2020.

Horizon Description	Particle Size %			Textural Class	Bulk Density
	Sand	Silt	Clay		(g cm^−3^)
Ap (0–30 cm)	45.2	44.3	10.5	Sandy Silt Loam	1.47
B1 (30–50 cm)	88.0	7.4	4.6	Sand	1.59
B2 (50–78 cm)	81.9	11.4	6.7	Loamy Sand	1.64
2C(k) (78–105 cm)	90.9	5.9	3.2	Sand	1.76
2C (105–120 cm)	95.4	2.9	1.7	Sand	1.66

Note: Explanations of horizons are presented in Table 3, and sand was defined between 2000 and 63 µm according to WRB.

## Data Availability

The original contributions presented in this study are included in the article/Appendix A. Further inquiries can be directed to the corresponding author.

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
