# Peer review of "Modeling Study on Optimizing Water and Nitrogen Management for Barley in Marginal Soils"

_plants, 2025, doi:10.3390/plants14050704_

Round 1
Reviewer 1 Report
Comments and Suggestions for Authors
This study provided an intreasting research that the agroecosystem model AgroC was calibrated and validated using a multiyear barley with field experiment under varying
Fertilization (N) and weather conditions (water stress). Generally, this paper is vell organized, but some contents could be further improved. The comments are as follows:
Title: It canbe simplified as “A Modelling Filed Study for Optimizing Water and Nitrogen Management for Barley in Marginal Soils”
“Introduction” and “Results and discussion”parts: Please provide more informations about the interaction of water stress and N fertilization on plant growth and yeild.
Reviewer 2 Report
Comments and Suggestions for Authors
General Comments:
The text is well-written, although some sections are slightly wordy. However, this is attributed to the writing style. Overall, the manuscript is clearly presented, and with minor revisions, it can be accepted. Since I am not a native English speaker, I did not comment on language aspects.
Abstract:
The first sentence is too general. It should serve as an introduction that highlights the issue of water and nitrogen fertilizers.
It is recommended to continue the text with the sentence about marginal soils.
Lines 22-24: The sentence is unclear—does the yield loss due to water and nitrogen deficiency occur cumulatively or separately?
Introduction:
The introduction provides sufficient background on barley, the AgroC model, and marginal soils.
Lines 71-72: Verify the percentages of barley production in Northern, Southern, and Western Europe. Three regions are mentioned, but only two values are provided.
Include a sentence on the importance of nitrogen in barley production on marginal soils.
The objectives are clearly stated.
Results and Discussion:
The results and discussion are clearly written, well-connected to previous findings, and have practical applications.
Comments:
Table: Yield – Provide an explanation since the results are presented first, but the abbreviations are introduced later in the Materials and Methods section.
Lines 194-219: While the results are well-presented, there is limited interpretation of their implications. For example, what do the differences in SWC and LAI between calibration and validation periods mean for the model's reliability or for barley production in marginal soils?
Materials and Methods:
Lines 648-649: It is stated that "only 13.4 m² (1.68 × 8 m) of the inner parts were harvested," while in Line 675, it is mentioned that "at barley physiological maturity (BBCH 92-99), the entire experimental plots (3 × 10 m = 30 m²) were harvested." Please clarify this discrepancy.
Reviewer 3 Report
Comments and Suggestions for Authors
The manuscript entitled “Optimizing Water and Nitrogen Management to Enhance Barley Yields on Marginal Soils: A Modelling Study” is an interesting work combining field work and modelling in order to develop strategies for reducing yield gaps in poor sandy soils in Baltic region.
This is an interesting and valuable work that fits the scope of Plants. However, before acceptance I advise the authors to give a better explanation and justification about the truly useful of this agroecosystem model AgroC. As you said “the model underestimated N uptake by grains” and you end up saying that general recommendation of N fertilisation is complicated due to yield variation between years. This means that apparently the strategy for reducing yield gaps is rather difficult accompanied by environmental costs, which makes a less valuable theoretical exercise as you proposed, and far away of your full objectives.
Minor aspects:
- Since N was added in the form of ammonium nitrate did you never study and observe ammonia loss?
- Your field treatments were based only in two treatments: N0 fertilization and N100, “as these treatments provide the most contrasting nutrient conditions for this field experiment”. How could you simulate the model to give you “a five- years spin-up prior the year 2020 to account for changes in soil N status due to the different fertilization rates”? How accurate could this be for you to reach values of N100-N120 as final recommendations ?
- “The highest nitrate leaching generally occurred during the winter months, where downward water flow occurs due to higher precipitation, low evaporation, and a lack of N uptake by plants.” However precipitation was higher in Summer months which does not follow your explanation. When we compared with dry matter evolution that leaching follow better the decline of vegetative stage. How do you explain this?
Reviewer 4 Report
Comments and Suggestions for Authors
This study evaluated the effects of N and water stress on barley grown in marginal soils using the AgroC model based on the three-years’ filed experimental data, which is an interesting topic. Here are some suggestions for improving the manuscript.
- The recommended amount of N fertilizer ambiguous in the manuscript. In the Abstract, it was mentioned that “Increasing N can still enhance yield, but optimal rates depend on climatic conditions, leading to uncertainty in fertilization” However, in the Conclusions, the optimal and economically beneficial N amount was determined as “between 100-120 kg ha-1”.
- The basis of determining the calibration period (2020 and 2022) and validation period (2021).
- The tables and figures in the manuscript should be more self-explanatory. In table 1, the meaning of “SWC”, “d”, “RESW”, and “BIAS”should be explained in the Note.
- The contentsof the Conclusions needs to be condensed, especially the first paragraph. Also, apart from “the efficiency of the fertilizer decreased” as the N fertilization rate increased, the nitrate leaching all increased (Figure 5).
Round 2
Reviewer 3 Report
Comments and Suggestions for Authors
You did answer to my questions and comments